# Interface-driven formation of a two-dimensional dodecagonal fullerene quasicrystal

M. Paßens[1], V. Caciuc[2], N. Atodiresei[2], M. Feuerbacher[3], M. Moors[1], R.E. Dunin-Borkowski[3], S. Blügel[2], R. Waser[1,4] & S. Karthäuser[1]

Since their discovery, quasicrystals have attracted continuous research interest due to their unique structural and physical properties. Recently, it was demonstrated that dodecagonal quasicrystals could be used as bandgap materials in next-generation photonic devices. However, a full understanding of the formation mechanism of quasicrystals is necessary to control their physical properties. Here we report the formation of a two-dimensional dodecagonal fullerene quasicrystal on a $Pt_3Ti(111)$ surface, which can be described in terms of a square–triangle tiling. Employing density functional theory calculations, we identify the complex adsorption energy landscape of the Pt-terminated $Pt_3Ti$ surface that is responsible for the quasicrystal formation. We demonstrate the presence of quasicrystal-specific phason strain, which provides the degree of freedom required to accommodate the quasicrystalline structure on the periodic substrate. Our results reveal detailed insight into an interface-driven formation mechanism and open the way to the creation of tailored fullerene quasicrystals with specific physical properties.

[1] Peter Grünberg Institut (PGI-7) and JARA-FIT, Forschungszentrum Jülich GmbH, 52425 Jülich, Germany. [2] Peter Grünberg Institut (PGI-1) and Institute for Advanced Simulation (IAS-1), Forschungszentrum Jülich GmbH, 52425 Jülich, Germany. [3] Peter Grünberg Institut (PGI-5) and Ernst Ruska-Centre for Microscopy and Spectroscopy with Electrons (ER-C), Forschungszentrum Jülich GmbH, 52425 Jülich, Germany. [4] IWE2 and JARA-FIT, RWTH Aachen University, 52056 Aachen, Germany. Correspondence and requests for materials should be addressed to N.A. (email: n.atodiresei@fz-juelich.de) or to M.F. (email: m.feuerbacher@fz-juelich.de) or to S.K. (email: s.karthaeuser@fz-juelich.de).

Quasicrystals (QCs) lack real-space periodicity and possess symmetry axes that are forbidden in periodic crystals. They exhibit rotational symmetry axes, such as fivefold, eightfold, tenfold and twelvefold, which are eponymous for the corresponding class of QCs[1–3]. For all classes of QCs, related periodic crystals are known, which are close in composition, have similar local atomic configurations and display diffraction patterns closely resembling those of QCs. Since these crystals approximate QCs in terms of structure and often also physical properties[4,5] they are referred to as approximants. For a single class of QCs usually diverse types of approximants with different lattice constants exist.

In the last decades, several dodecagonal QCs, as discussed here, were reported[6]. It was also demonstrated that dodecagonal photonic QCs form complete bandgaps (non-directional and for any polarization)[7,8]. As the order of symmetry of the crystal lattice increases, the Brillouin zone becomes more circular and a complete bandgap results. This property makes QCs that have a high degree of rotational symmetry highly promising for applications in diverse optical devices. Bulk metallic dodecagonal QCs have been observed in systems that include Ni–Cr[9], V–Ni(–Si)[10], Ta–Te[11] and a Mn-based alloy[12], while mesoscale dodecagonal QCs in soft-matter systems have been found in dendrimer liquid crystals[13], ABC star terpolymers[14] and surfactant-coated metallic nanoparticles[15,16].

The formation of an ultrathin two-dimensional (2D) quasi-crystalline $BaTiO_3$ film has been reported on Pt(111)[17], which is caused by a mismatch between a cubic and a hexagonal crystal structure. Recently, the self-assembly of molecular 2D QCs driven by hydrogen bonding[18] or metal-organic network formation[19] and the templated growth of fullerenes on Al-based QCs has been described[20,21]. Fullerenes deposited on (111) oriented surfaces of Au, Ag, Cu, Pb and Pt typically form commensurable, highly ordered, hexagonal, self-assembled monolayers[22–24]. It is well known that a $(\sqrt{13} \times \sqrt{13})R13.9°$ superstructure of fullerenes forms at room temperature on Pt(111) surfaces, while a slightly denser $(2\sqrt{3} \times 2\sqrt{3})R30°$ superstructure appears after annealing[25,26].

In the present study we report on fullerenes ($C_{60}$) adsorbed on a $Pt_3Ti(111)$ single crystal, intentionally terminated by two layers of Pt[27]. The resulting fullerene monolayer is characterized by low-energy electron diffraction (LEED) and scanning tunnelling microscopy (STM) supported by density functional theory (DFT) calculations. We will show that rather unforeseen domains of apparently dodecagonal fullerene QCs emerge and discuss the formation mechanism.

## Results

**Characterization of the fullerene monolayer.** Self-assembly of fullerenes at 320 K on a $Pt_3Ti(111)$ single crystal terminated by two layers of Pt ($2Pt–Pt_3Ti(111)$) at 320 K results in a monolayer with non-periodic and hexagonal domains. The analysis of four, large-scale STM images revealed that on average the non-periodic domains cover 60–70% of the surface area.

The non-periodic domains exhibit 12-fold symmetry, as verified by LEED and STM (Fig. 1, Supplementary Figs 1 and 2). The domains at the same time display an absence of real-space periodicity and the presence of a discrete diffraction pattern and therewith fulfil the necessary and sufficient conditions for being a QC[1,2]. The rotational symmetry of the diffraction pattern is 12-fold, and therefore, we interpret these non-periodic domains as 2D dodecagonal QCs. This is a highly unexpected result, since, as described above, fullerenes are known to form periodic superstructures on (111) surfaces of various metals including Pt[22–24].

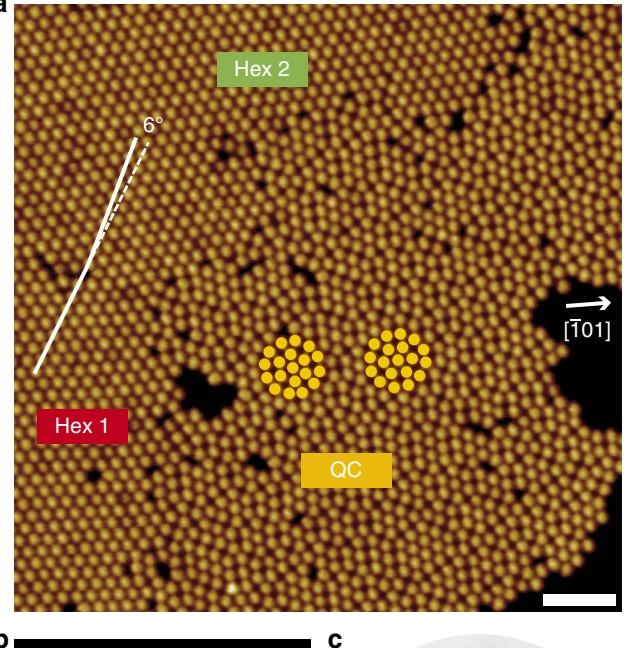

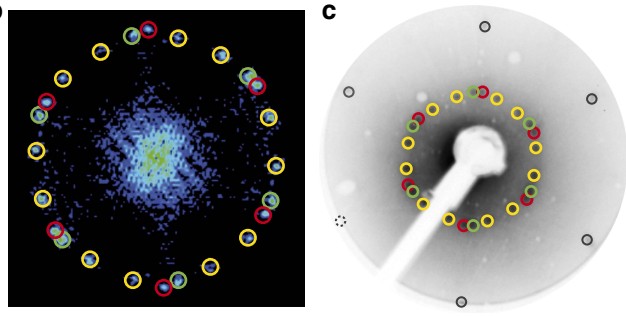

**Figure 1 | Analysis of a self-assembled monolayer of fullerenes on 2Pt–Pt₃Ti(111). (a)** Low-temperature UHV-STM image of a fullerene monolayer deposited on a $Pt_3Ti(111)$ single crystal terminated by two layers of Pt ($2Pt–Pt_3Ti(111)$) showing two differently oriented hexagonal (Hex1, Hex2) domains and one QC domain. Within the latter domain, two dodecagons with their inner hexagons rotated by ∼30° are marked. The indicated orientation of the substrate is determined directly from the atomically resolved $2Pt–Pt_3Ti(111)$ surface (scale bar, 5 nm; $U_{set} = -2.03$ V, $I_{set} = 0.47$ nA, 77 K). **(b,c)** FFT of the STM image and LEED pattern (energy: 19.5 eV), respectively, showing the spots of the two hexagonal domains (red and green circles) and the quasicrystalline domain with 12-fold symmetry (yellow circles).

The surface of a $2Pt–Pt_3Ti(111)$ single crystal is purely Pt terminated and the Pt–Pt distance differs by only $\Delta = 0.54\%$ with respect to Pt(111). This distance change is insignificant and cannot account for the different superstructures of fullerenes on Pt(111) and $2Pt–Pt_3Ti(111)$. We can therefore rule out incommensurability between the distance of adsorption sites provided by the metal alloy and the preferred fullerene–fullerene distance as a reason for QC formation. Instead, the origin of the observed 2D quasicrystalline structure can be assigned to interfacial interactions between the self-assembled monolayer of fullerenes and the $2Pt–Pt_3Ti(111)$ surface, both of which otherwise form periodic structures[23–27]. This behaviour is strikingly different from that associated with the templated growth of fullerenes[20,21], the self-assembly of molecular 2D quasicrystallites driven by hydrogen bonding[18] or the metal-organic network formation[19].

The 2D dodecagonal structure formed by fullerenes can be represented by a square–triangle tiling[11], as indicated by local

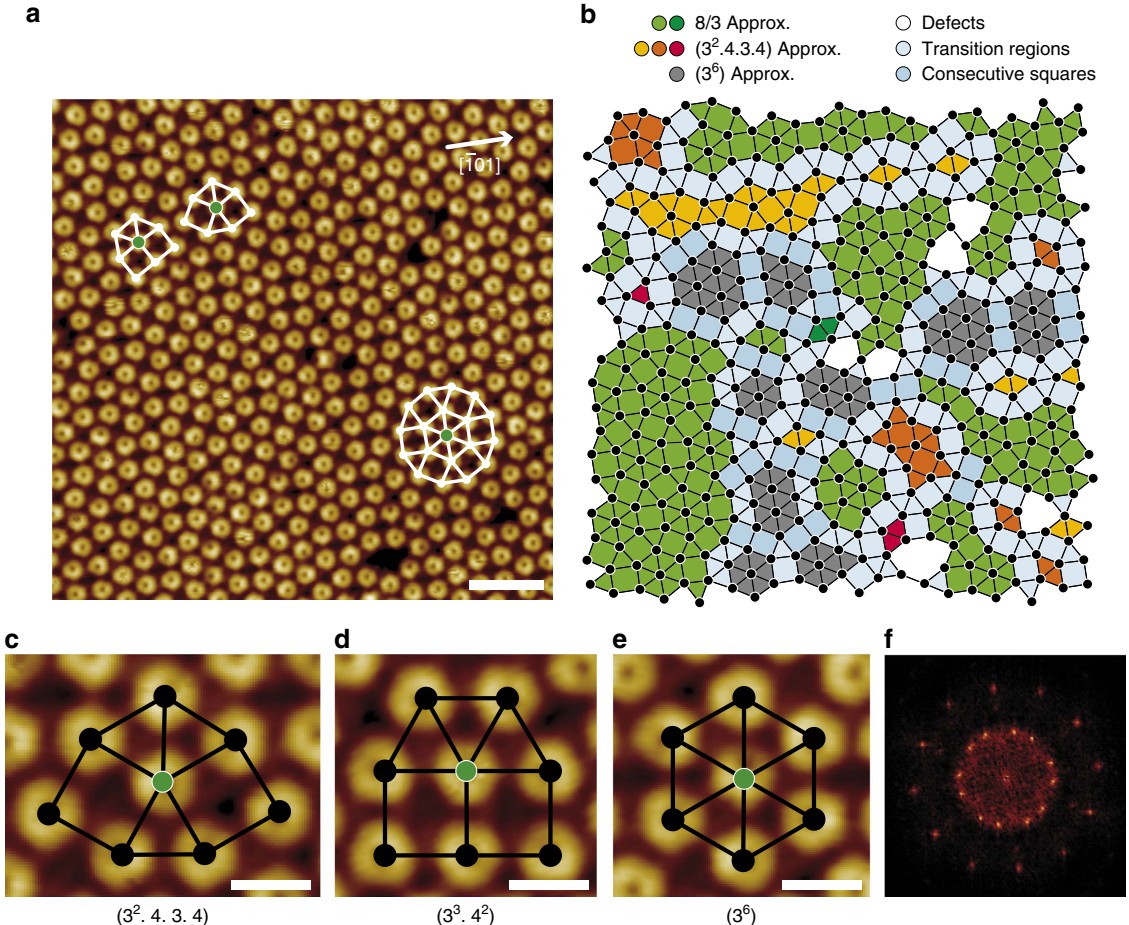

**Figure 2 | Dodecagonal square-triangle tiling measured by STM.** (**a**) High-resolution UHV-STM image of $C_{60}$ on 2Pt–Pt$_3$Ti(111) (scale bar, 3 nm; $U_{set} = -2.03$ V, $I_{set} = 0.47$ nA, 77 K). The HOMO of fullerenes facing the surface with a hexagon is imaged under these conditions and the characteristic dip in the middle of the HOMO is clearly resolved. One dodecagon and the local structures reproduced in **c,d** are indicated in white. (**b**) Square–triangle tiling extracted from **a**, with colour-coded decomposition into different types of approximants, see text. (**c–e**) Three types of local vertex configurations around one fullerene (marked green) (scale bar, 1 nm): (**c**) (3$^2$.4.3.4), (**d**) (3$^3$.4$^2$), (**e**) (3$^6$). (**f**) FFT of the STM image with 12-fold symmetry.

tiling patches in Fig. 2a. A tiling representation of the full area of Fig. 2a is given in Fig. 2b. A fast Fourier transform (FFT) of the STM image (Fig. 2f) displays two rings of 12 peaks, demonstrating the presence of dodecagonal rotational order. The mean intermolecular distance between the fullerenes in the 2D QC structure (Fig. 2) is $1.04 \pm 0.03$ nm, which is slightly larger than the intermolecular distance of 1.00 nm in a bulk crystal[28] and indicates that intermolecular van der Waals interactions are not the dominant driving force in the formation of the 2D QC structure. Local structural units of the square–triangle tiling are characterized by their vertex configuration, which can be represented by the Schläfli symbols $n^m$, where n denotes the kind and m the multiplicity of a polygon at a given vertex. In Fig. 2, vertex configurations corresponding to the three Schläfli symbols 3$^2$.4.3.4, 3$^3$.4$^2$ and 3$^6$ are identified, in addition to interpenetrating and fused dodecagons. We do not find a higher-order tiling, that is, a larger tiling that is inflated by a factor of $(2+\sqrt{3})$, and thus conclude that the present dodecagonal QC corresponds to a random tiling[1,13]. The tiling contains a small number of defects, which are polygons with interior angles of 60°, 90°, 120° and 150° consistent with the square–triangle tiling and indicates that the QC is not in its ground state[29].

The dodecagonal QC can be decomposed into different local areas corresponding to different types of approximants. For the dodecagonal QC, several approximants are known[14], and the colour coding in Fig. 2b demonstrates the presence of local areas of 8/3 approximant (37%), 3$^2$.4.3.4 approximant (8%) and 3$^6$ approximant (10%), as well as some defects (4%) and consecutive squares (11%). Boundaries that can be assigned to either of two neighbouring approximants are referred to as transition regions (30%). The coloured regions in Fig. 2b exhibit that the different approximants are randomly distributed.

This decomposition of the QC into local approximants is very similar to approaches taken, for example, for three-component polymer systems[14], tantalum-rich telluride[11], mesoporous silica[30] and for Mn–Cr–Ni–Si alloys[12]. In all these studies, the local domains investigated are decomposed into approximants of different type and orientation, but the ensemble of approximants as a whole is interpreted as a dodecagonal quasicrystal. This is in accordance with the general notion that the local occurrence of approximants of different types is a characteristic and generic property of any QC[6]. Clearly, this conception becomes less meaningful if the QC becomes small, particularly if phason strain is present, which in terms of approximant decomposition means that one type of approximant prevails. Our approach and interpretation is thus consistent with diverse literature examples, but nevertheless may be taken as an occasion to trigger a debate if the definition of a lower limit of spatial extension of a QC would be appropriate.

Hexagons ($3^6$) with two different orientations and a relative rotation of 30° are generally possible within a 12-fold symmetry (Fig. 1). However, we find that one orientation is distinctly preferred. This disparity has been observed before for other dodecagonal tilings although to a much smaller extent[11]. In our images, tiling statistics (Supplementary Fig. 3 and Supplementary Table 1) reveal the presence of tile edges along $<1-21>$ directions of the underlying 2Pt–Pt$_3$Ti(111) substrate roughly twice as often as edges along $<-101>$ directions. Another important number that can be used to characterize a quasiperiodic square–triangle tiling is the triangle/square number ratio, which is $R_{tr/sq} = (4/\sqrt{3}) = 2.309$ for deterministic or maximally random tilings[12]. Remarkably, we obtain a value of $R_{tr/sq} = 2.67$ (Supplementary Fig. 3 and Supplementary Table 1). This enhanced value of $R_{tr/sq}$ and the preferred alignment of tile edges observed along $<1-21>$ directions indicate an inequality in the local structural units within the dodecagonal tiling in contrast to, for example, soft QCs[14]. This directional inequality is an immediate consequence of the interface-driven formation of the fullerene 2D QC, since the crystallographic directions $<1-21>$ and $<-101>$ of the underlying hexagonal surface are not equivalent (Supplementary Fig. 4).

**Phason-strain analysis.** Careful inspection of the FFT (Fig. 2f) reveals that the 12 spots are not in the positions that correspond to an ideal undistorted dodecagonal QC, but exhibit slight spot shifts. These spot shifts are more pronounced for the inner ring of the FFT, where they lead to an arrangement of pairs of spots rather than to a radially even distribution. This specific pattern of shifts qualitatively corresponds to the presence of $\Gamma^{1b}$ phason strain[31]. Phason strain is characteristic of quasicrystalline structures and describes local deviations from ideal quasicrystalline order. Uniform phason strain in dodecagonal QCs is described by a $2 \times 2$ matrix, the phason matrix, which can be determined by measuring the 12 positions of spots in a ring in the diffraction pattern or FFT[32]. Following a procedure provided by Leung et al.[32], we determined the positions of the 12 {1 1 0 0} type spots in FFTs of STM images taken in six different sample areas. On average, we obtain a phason matrix

$$\mathbf{A} = a \begin{pmatrix} 1 & 0.1 \\ -0.1 & -1 \end{pmatrix},$$ where $a \approx 1.3$. This result demonstrates

that considerable phason strain is present in the structure.

The form of the matrix thus corresponds to a matrix describing $\Gamma^{1b}$ phason strain[31] and also the entries are close, taking the measurement error into account. As demonstrated by Socolar et al.[31] and Ishii[33], phason strain mediates a continuous transition from ideal QCs to low-order approximants, down to very small lattice constants. The phason strain detected can be understood as a mechanism to accommodate the 2D QC to the periodic hexagonal substrate. Figure 2b demonstrates a slight prevalence of the 8/3 approximant over the other types present within the surface area investigated. In addition, the $3^6$ approximant is present in a considerable amount. This is consistent with the observation of phason strain[6] and a deviation of the triangle/square ratio from the value of ideal dodecagonal QCs.

**Adsorption energy landscape of 2Pt–Pt$_3$Ti(111).** Since, as discussed above, a direct geometrical origin for the formation of dodecahedral symmetry can be excluded, we concentrate here on

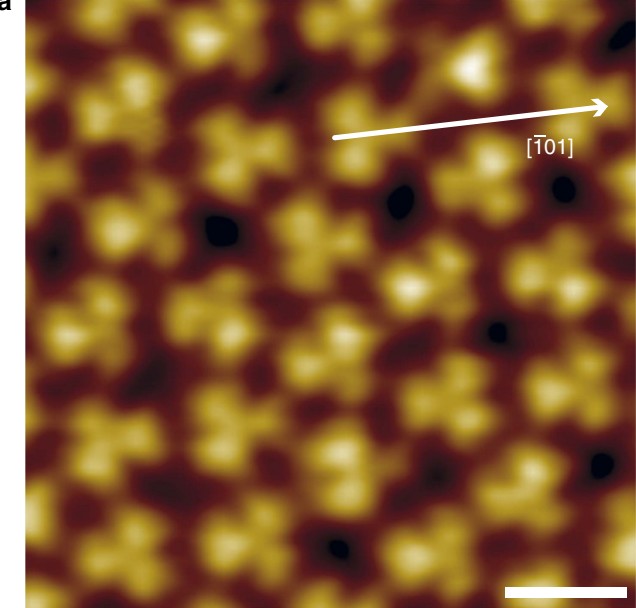

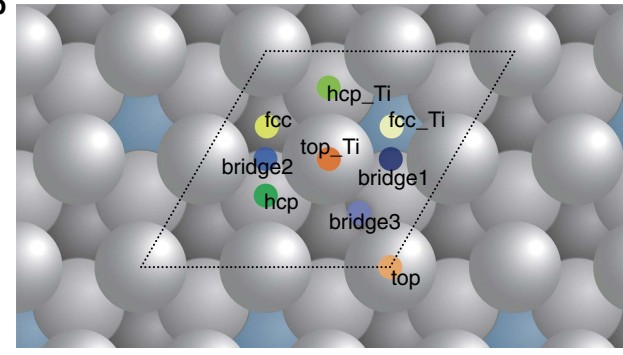

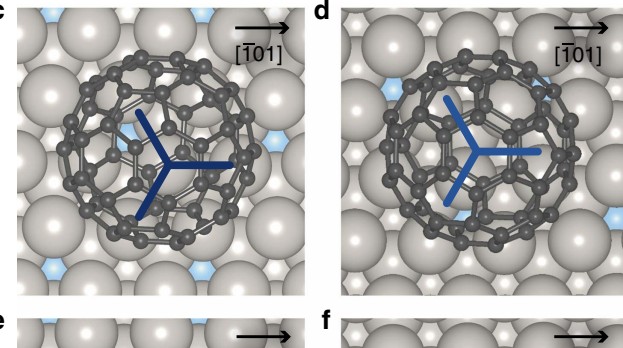

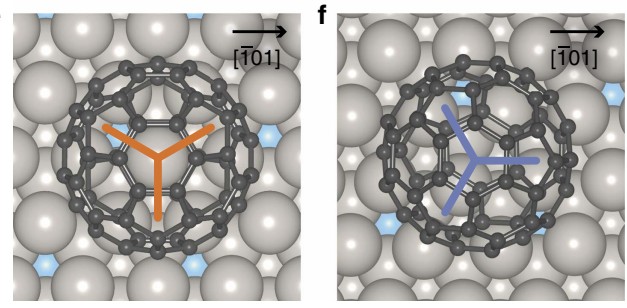

**Figure 3 | Adsorption configuration of fullerenes on 2Pt–Pt$_3$Ti(111).**
(**a**) High-resolution UHV STM image of C$_{60}$ on 2Pt–Pt$_3$Ti(111) (scale bar, 1 nm; $U_{set} = +2.22$ V, $I_{set} = 2.9$ nA, 77 K, slightly low pass filtered). The three-lobe structure of the unoccupied molecular orbitals of fullerenes facing the surface with a hexagon is clearly visible. (**b**) Schematic diagram indicating nine positions in the ($1 \times 1$) 2Pt–Pt$_3$Ti(111) surface unit cell considered for the adsorption of fullerenes in DFT calculations using a ($3 \times 3$) supercell (Supplementary Fig. 5). (**c**–**f**) Schematic top view of a fullerene adsorbed on 2Pt–Pt$_3$Ti(111) in bridge1, bridge2, top_Ti and bridge3 positions, respectively.

**Table 1 | Adsorption energies and Pt-C bond lengths.**

|  | top | top_Ti | hcp | hcp_Ti | bridge1 | bridge2 | bridge3 | fcc | fcc_Ti |
|---|---|---|---|---|---|---|---|---|---|
| $E_{ads}$, eV | − 4.261 | − 4.583 | − 4.487 | − 4.343 | − 4.796 | − 4.754 | − 4.543 | − 4.123 | − 3.952 |
| $L_{Pt-C}$, nm | 0.216 | 0.215 | 0.215 | 0.214 | 0.208 | 0.211 | 0.212 | 0.215 | 0.215 |

The adsorption energies ($E_{ads}$) and the Pt-C bond lengths ($L_{Pt-C}$, nm) have been calculated for nine configurations of fullerenes on a 2Pt–Pt$_3$Ti(111) surface. The shortest $L_{Pt-C}$ value obtained for the respective adsorption position is given in the table. The resulting $L_{Pt-C}$ values correspond approximately to the sum of the covalent radii of Pt and C (0.212 nm) and, thus, point to a strong interaction between the fullerene molecule and the 2Pt–Pt$_3$Ti(111) surface.

the energetic landscape provided by the 2Pt–Pt$_3$Ti(111) surface. In detail, we analyse adsorption energies resulting from the deposition of fullerenes on 2Pt–Pt$_3$Ti(111). Our high-resolution STM investigations (Fig. 3) consistently reveal that all fullerenes exhibit a three-lobe orbital structure[24], which is aligned along the < −1 0 1 > directions of the underlying substrate. The apparent height variations are below ± 0.02 nm, resulting from small tilts of the molecules. We focus on DFT simulations of single fullerenes that are adsorbed with a hexagon parallel to the 2Pt–Pt$_3$Ti(111) surface. Specifically, we consider nine adsorption positions (Table 1, Fig. 3, Supplementary Figs 5 and 6) to map the adsorption energy on the complex surface adequately.

The resulting adsorption energies and Pt–C bond lengths are within the range of values characteristic for the adsorption of fullerenes on Pt(111)[26] and are consistent with a covalent bond character. A remarkable result is that the Ti atoms beneath the two Pt overlayers affect the adsorption energies on otherwise similar positions on the surface. That is, the top and top_Ti, hcp and hcp_Ti, as well as fcc and fcc_Ti positions are distinctly different in energy. Moreover, three different bridge positions can be distinguished on the 2Pt–Pt$_3$Ti(111) surface, while only one exists for Pt(111). The calculations predict that two bridge configurations, bridge1 and bridge2, are energetically most favourable, followed by the top_Ti configuration (Fig. 3 and Table 1). The adsorption energies for all other configurations are smaller. We therefore assume that the fullerenes adopt adsorption configuration bridge1 or bridge2.

According to our DFT calculations, fullerenes that are adsorbed in bridge and non-bridge configurations adopt different orientations (Fig. 3). While the three-lobe orbital structure of fullerenes is aligned along the < −1 0 1 > directions of the underlying substrate in each bridge configuration, it is rotated by 30° in all other investigated configurations (Supplementary Fig. 6). Moreover, the molecules are slightly tilted in the bridge configurations after relaxation, whereas they adsorb with the hexagon parallel to the 2Pt–Pt$_3$Ti(111) surface in all other configurations. These results provide confirmation that the fullerenes that make up the dodecagonal structure and display a slightly tilted three-lobe orbital structure oriented along the < −1 0 1 > directions occupy bridge1 or bridge2 positions.

The energetically favoured bridge1 position is threefold degenerate, indicating that fullerenes have a degree of freedom to adjust their nearest neighbour distance (Fig. 4). Another point is that fullerenes can be tilted easily and, thus, are able to fine-tune their intermolecular interactions. In additional, our DFT calculations indicate that the Pt surface layer is relaxed under the influence of fullerene adsorption. All of these interfacial processes contribute to the formation of a fullerene quasicrystal.

The schematic diagram shown in Fig. 4b illustrates the adaptation of the local 3$^2$.4.3.4 structure, which is typical for our 2D dodecagonal QC, to the energetically favourable bridge1 positions on the 2Pt–Pt$_3$Ti(111) surface. However, a ($\sqrt{13} \times \sqrt{13}$)R13.9° superstructure of fullerenes, as is commonly observed on the Pt(111) surface, which exhibits only one kind of bridge positions, can be realized on the 2Pt–Pt$_3$Ti(111) surface only if alternatively bridge1 and other less favoured bridge2

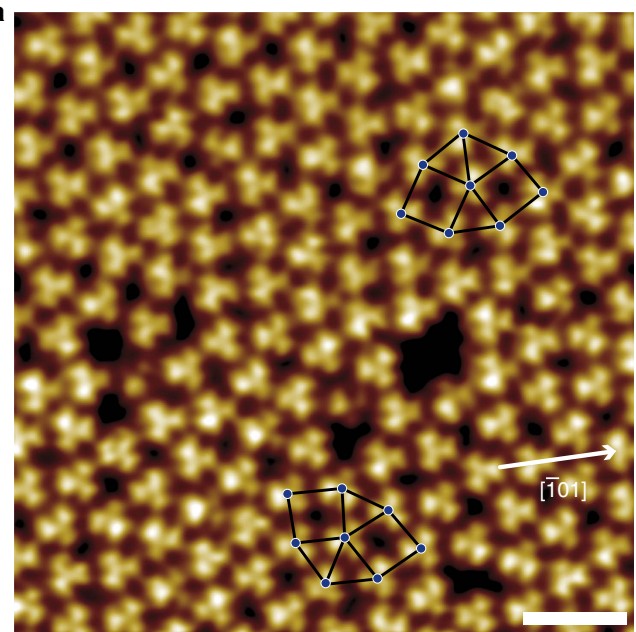

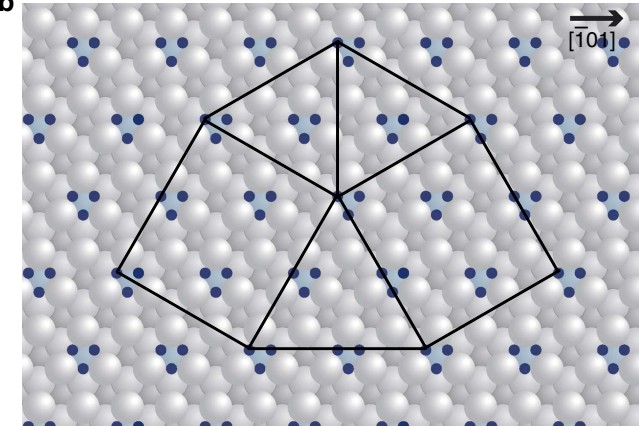

**Figure 4 | Proposed adsorption model of fullerenes on 2Pt–Pt$_3$Ti(111).** (**a**) High-resolution UHV STM image of C$_{60}$ on 2Pt–Pt$_3$Ti(111) (scale bar, 2 nm; $U_{set} = + 2.22$ V, $I_{set} = 2.9$ nA, 77 K, slightly low pass filtered). (**b**) Schematic diagram showing the local 3$^2$.4.3.4 structure assuming adsorption only on bridge1 positions of an ideal 2Pt–Pt$_3$Ti(111) surface.

and bridge3 positions are occupied. The absence of the ($\sqrt{13} \times \sqrt{13}$)R13.9° superstructure demonstrates how the adsorption energy landscape, which is influenced on the 2Pt–Pt$_3$Ti(111) surface by third-layer atoms, is critically responsible for the respective monolayer structures of organic molecules.

**General aspects of the fullerene QC structure.** A more general applicability of this concept was demonstrated by depositing fullerenes on Pt$_3$Ti(111) single crystals terminated by a single Pt

layer (see Supplementary Fig. 7). Based on geometrical considerations, three different bridge sites, with respect to subsurface Ti atoms, can be identified on the Pt–Pt$_3$Ti(111) surface, too. Here the subsurface Ti atoms should influence the local adsorption energy of fullerenes even more than in the case of the 2Pt–Pt$_3$Ti(111) surface since the surface Pt atoms are in direct interaction with the subsurface Ti atoms. This should lead to a considerable energetic discrimination of the three different bridge positions, which in turn results in the formation of a 2D dodecagonal QC, like shown in Supplementary Fig. 7.

In principle, the full range of Pt$_3$M alloys, where M is a three-dimensional or four-dimensional transition metal, is suitable for this route to QC formation, with the strength of the fullerene–metal interaction depending on the kinds of transition metal and overlayers used. Interestingly, we find that an additionally deposited second fullerene layer also adopts a quasicrystalline structure. This behaviour opens up the possibility to create a QC multilayer or even a three-dimensional fullerene QC.

## Discussion

Here we presented the formation of a nanoscale 2D fullerene monolayer, which does not show long-range periodicity, and the LEED pattern of which displays 12-fold rotational symmetry. The fullerene monolayer can thus be interpreted as a 2D dodecagonal QC, which is composed of an organic molecule adsorbed on a periodic metallic surface. As such, it does not belong to any previously known class of dodecagonal QCs. Our investigations, on the other hand, demonstrate that even though the structure of the fullerene monolayer fulfils the formal conditions defining a quasicrystal, it challenges the understanding of quasicrystallinity for structures on small length scales, particularly if these are imperfect and/or contain phason strain.

In conclusion, we report the formation of a nanoscale dodecagonal QC with a triangle/square tiling by the self-assembly of fullerenes on a 2Pt–Pt$_3$Ti(111) surface. Most interestingly, the driving force for the 2D QC structure is a substrate-related adsorption energy landscape. By employing DFT calculations, we have identified the preferred adsorption sites of the fullerenes, which are determined critically by Ti atoms located in the third layer of the 2Pt–Pt$_3$Ti surface. These Ti atoms induce a complex adsorption energy landscape, which results, for example, in three distinct bridge positions that are experienced by the fullerenes. In addition, we demonstrate the presence of QC-specific phason strain, which provides the degree of freedom that is required to accommodate the quasicrystalline structure on the periodic substrate. This concept, which results in the ability to generate QCs by tuning an adsorption energy landscape by the introduction of suitable atoms in subsurface layers of a periodic substrate, can be applied to other organic molecules, such as endo-fullerenes and functionalized fullerenes. Our results have significant implications for the directed synthesis of a wide variety of quasicrystalline structures, whose physical properties are tunable by the choice of substrate or adsorbed molecules.

## Methods

**Sample preparation and STM measurements.** A Pt$_3$Ti(111) single crystal (MaTecK, Germany) was prepared by several cycles of ion sputtering ($3 \times 10^{-5}$ mbar neon pressure at 1 keV for 10 min) and annealing (at 1,200 K for 25 min) under ultrahigh vacuum (UHV) conditions, leading to the presence of two platinum overlayers on the Pt$_3$Ti(111) surface[27]. C$_{60}$ molecules (Sigma Aldrich, purity 99.9%) were deposited by sublimation at 320 °C using a Knudsen cell with a deposition rate of 0.04 ML per min, while the Pt$_3$Ti(111) substrate was heated to 50 °C. The sample was then transferred into a commercial Createc LT UHV STM ($<1 \times 10^{-10}$ mbar) and characterized at 77 K. STM measurements were performed in constant current mode using electrochemically etched tungsten tips.

**LEED measurements.** LEED images were acquired using a filament current of 2.55 A and a screen voltage of 3 keV was used. To avoid beam-induced surface damage, the effective electron beam voltage was kept below 200 nA during all measurements. The OCI BDL8000IR-MCP LEED system was used to amplify the diffracted electrons from the surface via two microchannel plates for imaging the diffraction pattern while the sample was kept at room temperature.

**DFT calculations.** DFT[34] calculations were performed using the Perdew–Burke–Ernzerhof exchange correlation functional[35] for pseudopotentials obtained with the projector augmented wave method[36], as implemented in the Vienna ab-initio simulation package (VASP) code[37–39]. DFT simulations were performed on single fullerenes adsorbed with a hexagon parallel to the 2Pt–Pt$_3$Ti(111) surface. The C$_{60}$ on 2Pt–Pt$_3$Ti(111) system was modelled in the form of a $(3 \times 3)$ slab consisting of five atomic layers using the Pt$_3$Ti bulk lattice parameter of 0.3949 nm with one C$_{60}$ molecule adsorbed on the Pt surface. For an energy cutoff of 500 eV, the molecule surface ground-state geometry was obtained by relaxing the C$_{60}$ and the two surface layers until the calculated forces were smaller than 1 meV Å$^{-1}$.

**Data availability.** The authors declare that the data supporting the findings of this study are available within the article and its Supplementary Information file or from the corresponding authors on reasonable request.

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

## Acknowledgements

M.P. and S.K. gratefully acknowledge the help of Jochen Friedrich, Stephan Masberg and Thomas Pössinger. Computations were carried out using the high-performance computer JUQUEEN operated by the Jülich Supercomputing Centre at the Forschungszentrum Jülich GmbH. V.C., N.A. and S.K. gratefully acknowledge financial support from the Volkswagen-Stiftung through the 'Optically Controlled Spin Logic' project.

## Author contributions

M.P. prepared the single crystal surface termination and the fullerene layer, performed the low-temperature STM and LEED investigations and made the initial experimental discovery. M.M. conducted the single crystal preparation and assisted the LEED measurements. M.P. and S.K. designed the conceptual approach. V.C. and N.A. performed the DFT calculations. M.F. performed the phason-strain analysis. M.P., V.C., N.A., M.F. and S.K. analysed the results. S.K., M.F. and M.P. wrote the manuscript with input from all co-authors. S.K. coordinated the efforts of the research team. All authors discussed the results and improved the manuscript.

## Additional information

**Competing interests:** The authors declare no competing financial interests.

