## [Peer Review File · Nature Communications]

Editorial Note: Parts of this peer review file have been redacted as we could not obtain permission to publish the reports of Reviewer 3.

Reviewers' comments:

Reviewer #1 (Remarks to the Author):

The manuscript entitled "Interface-driven formation of a two-dimensional dodecagonal fullerene quasicrystal" by M. Paßens et al. reports on the formation of a two-dimensional dodecagonal quasicrystal that is composed of fullerenes on a Pt₃Ti(111) surface. The experimental observations are based on in-situ scanning tunneling microscopy (STM) and low-energy electron diffraction (LEED) and are supported by DFT calculations. In combination of all three techniques, the authors conclude in an interface-driven quasicrystal formation with quasicrystal-specific phason strain.

On the experimental side, the analysis is demanding due to presence of structural disorder (besides any possible phason strain) and the coexistence of periodic structures. However, I will show in the following that the observed structure in LEED and well as by STM is NOT a dodecagonal quasicrystal. Instead, it is a periodic approximant structure based on a triangle-square tiling with local disorder. As the authors point out, the central element is a dodecagon with their inner hexagons rotated by approximately 30°. This wagon-wheel structure consists of a triangle and square tiling with local 3².4.3.4 and 3⁶ vertex structures, all common elements of dodecagonal quasicrystals. However, the long-range order, which leads to sharp electron diffraction spots and to distinct maxima in the Fourier transform of the STM data, belongs to a periodic (hexagonal) arrangement of the wagon-wheel element. In fact, such a hexagonal arrangement can explain straightforwardly all experimental observations without the need of phason strain.

The simple wagon-wheel element shown in Fig. 1(a) (see Attachment) can be arranged in a strictly hexagonal lattice as shown in Fig. 1(b). This leads to a diffraction pattern in (c) that is identical to the pattern reported in the manuscript. Note that this is a periodic approximant structure.

The direct comparison of the experimental diffraction pattern (see Fig. 2 below and Fig. 2(f) in the manuscript) with the hexagonal structure highlights the perfect matching. Note further that the experimentally observed triangle/square number ratio of 2.67 agrees accurately with the 8/3 ratio (24 triangles and 9 squares) of the hexagonal structure. Clearly, there is structural disorder present, which complicates structural assignments. However, the diffraction and FFT data show clearly the underlying periodic order. The observed wagon wheel motif is a common approximant structure that has been also reported for the 2D interface-driven oxide quasicrystal [1,2].

Based on these arguments, the manuscript does not provide evidence for quasicrystal formation nor on any phason strain. Nevertheless, the interesting observation of a periodic approximant structure does not rule out the existence of an aperiodic quasicrystal under slightly different preparation conditions. Furthermore, the fullerene-based structure formation might shine light on the mechanisms that lead to specific triangle-square tilings as discussed nicely in the manuscript. Therefore, I recommend to rewrite the manuscript with focus on approximant structures. With its present focus I do not recommend publication.

References:

[1] Quasicrystalline structure formation in a classical crystalline thin-film system

S. Förster, K. Meinel, R. Hammer, M. Trautmann, and W. Widdra
Nature 502, 215-8(2013).

[2] Ultrathin Perovskites: From Bulk Structures to New Interface Concepts

S. Förster and W. Widdra

in Oxide Materials at the Two-Dimensional Limit, F. P. Netzer, and A. Fortunelli eds. (Springer, Heidelberg, 2016).

http://link.springer.com/content/pdf/10.1007%2F978-3-319-28332-6_13.pdf

Reviewer #2 (Remarks to the Author):

The authors present high-quality STM and LEED data, and make a careful, quantitative, and well-referenced argument that the presence of sub-surface Ti substantially alters the adsorption energy of C60 on Pt3Ti-2Pt, and that this alteration is responsible for the formation of a quasicrystalline monolayer. It is an interesting result and the authors' analysis is convincing.

Taken on their own, the VASP calculations seem to be a weak point in this study. Some evidence should be provided to support the choice of the calculation cell size and the choice of degrees of freedom to relax. The lateral size of the cell seems small: especially if there is any significant relaxation of surface or subsurface atoms, the slab ought to extend well beyond the fullerene radius if the results are to be used quantitatively. Also, the bridge1 and bridge2 sites are close enough in their calculated adsorption energies that they should probably be regarded as qualitatively isoenergetic (instead of "most highly" and "less" favored) -- even if VASP's accuracy is assumed at this level (dubious), the energy separation is on the order of kT at the surface preparation temperature.

The authors indicate that a quasicrystalline monolayer also forms when Pt3Ti has only a single overlayer of platinum. This data should be included, either in the paper or the SI. If at all possible, the authors should repeat their experiment with a Pt(111) surface, as well, to verify that under the specific conditions used, they do not observe quasicrystalline ordering. Adsorption energy calculations for both of these surfaces (Pt3Ti-1 Pt and Pt(111)) should be done in a similar fashion to the calculations presented.

The authors' interesting and surprising conclusion in this study is that third-layer Ti atoms modify the energy landscape enough to drive the formation of a quasicrystalline monolayer. The strongest possible argument to support this conclusion would be to show that the adsorption energies are appropriately modified in all cases where quasicrystals are observed (Pt3Ti with 2 layers and 1 layer of Pt), and that they are not in adsorption systems (presumably, C60 on Pt(111)) where only hexagonal ordering is seen.

Response to the referees comments on

Manuscript: NCOMMS-16-16737-T

“Interface-driven formation of a two-dimensional dodecagonal fullerene quasicrystal”

We thank the referees for their time they investigated in our manuscript, the in depth analysis of our results, and the valuable comments that helped us to improve the manuscript. In the following we reply all questions raised by the referees and address all subjects of debate. The corresponding changes made are highlighted in the manuscript and quoted here.

Citations quoted as numbers in square brackets in this reply refer to the references in the manuscript. Additional citations used only here are quoted as [Rx]. Analogously we refer to figures in the manuscript and in the reply.

Referee: #1

The manuscript entitled “Interface-driven formation of a two-dimensional dodecagonal fullerene quasicrystal” by M. Paßens et al. reports on the formation of a two-dimensional dodecagonal quasicrystal that is composed of fullerenes on a Pt3Ti(111) surface. The experimental observations are based on in-situ scanning tunneling microscopy (STM) and low-energy electron diffraction (LEED) and are supported by DFT calculations. In combination of all three techniques, the authors conclude in an interface-driven quasicrystal formation with quasicrystal-specific phason strain.

On the experimental side, the analysis is demanding due to presence of structural disorder (besides any possible phason strain) and the coexistence of periodic structures. However, I will show in the following that the observed structure in LEED and well as by STM is NOT a dodecagonal quasicrystal. Instead, it is a periodic approximant structure based on a triangle-square tiling with local disorder. As the authors point out, the central element is a dodecagon with their inner hexagons rotated by approximately 30°. This wagon-wheel structure consists of a triangle and square tiling with local $3^2.4.3.4$ and 3^6 vertex structures, all common elements of dodecagonal quasicrystals. However, the long-range order, which leads to sharp electron diffraction spots and to distinct maxima in the Fourier transform of the STM data, belongs to a periodic (hexagonal) arrangement of the wagon-wheel element. In fact, such a hexagonal arrangement can explain straightforwardly all experimental observations without the need of phason strain.

The simple wagon-wheel element shown in Fig. 1(a) (see Attachment) can be arranged in a strictly hexagonal lattice as shown in Fig. 1(b). This leads to a diffraction pattern in (c) that is identical to the pattern reported in the manuscript. Note that this is a periodic approximant structure.

The direct comparison of the experimental diffraction pattern (see Fig. 2 below and Fig. 2(f) in the manuscript) with the hexagonal structure highlights the perfect matching. Note further that the experimentally observed triangle/square number ratio of 2.67 agrees accurately with the 8/3 ratio (24 triangles and 9 squares) of the hexagonal structure. Clearly, there is structural disorder present, which complicates structural assignments. However, the diffraction and FFT data show clearly the underlying periodic order. The observed wagon wheel motif is a common approximant structure that has been also reported for the 2D interface-driven oxide quasicrystal [1,2].

Based on these arguments, the manuscript does not provide evidence for quasicrystal formation nor on any phason strain. Nevertheless, the interesting observation of a periodic approximant structure does not rule out the existence of an aperiodic quasicrystal under slightly different preparation conditions. Furthermore, the fullerene-based structure formation might shine light on the mechanisms that lead to specific triangle-square tilings as discussed nicely in the manuscript. Therefore, I recommend to rewrite the manuscript with focus on approximant structures. With its present focus I do not recommend publication.

References:

[1] Quasicrystalline structure formation in a classical crystalline thin-film system

S. Förster, K. Meinel, R. Hammer, M. Trautmann, and W. Widdra, Nature 502, 215-8(2013).

[2] Ultrathin Perovskites: From Bulk Structures to New Interface Concepts

S. Förster and W. Widdra in Oxide Materials at the Two-Dimensional Limit, F. P. Netzer, and A. Fortunelli eds. (Springer, Heidelberg, 2016).

http://link.springer.com/content/pdf/10.1007%2F978-3-319-28332-6_13.pdf

Reply: The referee argues that the wagon-wheel structure is sufficient to explain the FFT of our experimental STM image and the triangle/square ratio. Indeed, the calculated diffraction pattern of the wagon-wheel structure resembles our FFT. However, the wagon-wheel structure is not the only structure that leads to such a diffraction pattern. There are diverse other structures leading to FFT patterns identical within the experimental resolution, one of which is the phason-strained dodecagonal quasicrystal we claim to have observed. Corroboratively, it has been demonstrated recently that the Fourier spectrum does not suffice to differentiate between a real-space dodecagonal quasicrystalline structure or a related periodic structure [R1].

In Fig. 2 of his report, the referee compares the calculated pattern of a hexagonal wagon-wheel arrangement with the FFT of our experimental STM image. The referee does so by using a grid superposed to the experimental pattern. The peaks in the FFT indeed correspond to crossing points of that grid. Upon careful inspection, however, it occurs that the grid used by the referee does not consist of equidistant lines as it should for a periodic pattern. It is clearly visible that the distances between the grid lines have been adjusted such that the crossing points match the peaks of the FFT. This procedure is highly suggestive and appears questionable to us. Using, on the other hand, a pattern of equidistant grid lines, a perfect match of all peaks of the FFT with crossing points is not possible (Fig. R1a).

Fig. R1: Referee's Figure 2 with overlaid equidistant hexagonal grid (blue) in (a). The equidistant grid obviously deviates from the non-equidistant grid (green) used by the referee.

Furthermore, an (equidistant!) hexagonal grid can be used to describe the peak maxima of Fig. 14a in the paper of Socolar (Ref [25]), with a similar matching agreement as to our FFT pattern (Fig. R2). Notably, the figure of Socolar is the diffraction pattern of a dodecagonal quasicrystal including phason strain, i.e. exactly the type of structure we claim to have observed in our experimental images.

Fig. R2: Equidistant hexagonal grid (blue) superposed to the diffraction pattern of a dodecagonal quasicrystal including phason strain (Fig. 14 of Socolar [25]).

We thus conclude that our experimental FFT pattern does neither allow to conclude on the presence of periodic hexagonal structure, nor rule out the presence of a phason-strained dodecagonal quasicrystal.

The referee states that the triangle/square ratio of our structure matches that of a hexagonal wagon-wheel arrangement. However, it has been demonstrated that the presence of a given triangle/square ratio does not allow to conclude on the real-space structure (which simply means that different structures can be constructed using a given ratio of triangle and square tiles). It has, for example, been demonstrated that random-Stampfli approximants at zero phason strain possess exactly the same triangle/square ratio of 2.309 [R1] as deterministic (i.e. ideal) and random dodecagonal tilings [9].

Finally, the referee's statement, saying that the periodic hexagonal wagon-wheel structure "can explain straightforwardly all experimental observations" is not true. Factually our real-space STM image (Fig. 2a) and its tiling (Fig. 2b) are in direct contradiction with the mere presence of a periodic hexagonal wagon-wheel structure. The structure in the experimental STM image is clearly not identical to the referee's Fig. 1b representing the wagon-wheel structure. The surface section in our experimental image is composed of small local areas, which individually can be identified as approximants of the ideal dodecagonal quasicrystal. It contains 8/3 areas, the approximant corresponding to the hexagonal periodic wagon-wheel structure, but it also contains areas of 3².4.3.4 approximant, hexagonal approximant, consecutive squares and diverse transition regions.

Generally, a quasicrystalline structure, even an ideal one, can always locally be decomposed into small areas which individually can be identified as characteristic tiling patches corresponding to approximants of different order [2]. Nevertheless, the total structure still remains a quasicrystal, unless the local approximant areas are periodically repeated on a nonlocal scale.

We also do not agree with the referee's statement that our data does not provide evidence for the presence of phason strain. Following the procedure described by Leung et al. [26] for the identification of phason strain in a dodecagonal quasicrystal, we have obtained the nonzero phason matrix $1.3 \begin{pmatrix} 1 & 0.1 \\ -0.1 & -1 \end{pmatrix}$. This phason matrix corresponds to the presence of Γ^{1b} phason strain in a dodecagonal quasicrystal [25]. Note furthermore that our matrix is significantly different from the phason matrix corresponding to an 8/3 approximant. The latter reads $0.26 \begin{pmatrix} 1 & -0.3 \\ 0.3 & 1 \end{pmatrix}$ [11], which further supports our statements above.

These points demonstrate that the structure observed experimentally can be described as a dodecagonal quasicrystal containing phason strain. The argumentation line of the referee, who states that the structure observed corresponds to a wagon-wheel structure or 8/3 approximant, is not conclusive and partly in conflict with the experimental facts.

Changes in the manuscript: Admittedly, we recognize that we have not sufficiently covered the subject from the approximant perspective in our manuscript. Therefore, we made the following changes:

- Figure 2: We have replaced Fig. 2b by a color-coded version in which the different local approximant areas are identified. The caption was changed accordingly.

- p.7: We have added a section introducing the concept of approximants and describing the decomposition of the investigated surface area into local approximants. We wrote:
 “The dodecagonal QC can be decomposed into different local areas corresponding to different types of approximants. Approximants are periodic crystals composed of local structural units that also occur in the QC. For a dodecagonal QC several approximants are known [11], and the color coding in Fig. 2b demonstrates the presence of local areas of $8/3$ approximant, $3^2.4.3.4$ approximant, and 3^6 approximant, as well as some defects and consecutive squares. Boundaries that can be assigned to either of two neighboring approximants are referred to as transition regions. The colored regions in Fig. 2b exhibit that the different approximants are randomly distributed.”
- p.8: We added a statement clarifying that the values of the matrix elements of the phason matrix demonstrate the presence of phason strain. We wrote:
 “This result demonstrates that considerable phason strain is present in the structure.”
- p.9: We added a statement substantiating the consistency of our experimental results:
- “Fig. 2b demonstrates a slight prevalence of the $8/3$ approximant over the other types present within the surface area investigated. In addition the 3^6 approximant is present in a considerable amount. This is consistent with the observation of phason strain [2] and a deviation of the triangle/square ratio deviating from the value of ideal dodecagonal QCs.”

Referee: #2

#2: The authors present high-quality STM and LEED data, and make a careful, quantitative, and well-referenced argument that the presence of sub-surface Ti substantially alters the adsorption energy of C₆₀ on Pt₃Ti-2Pt, and that this alteration is responsible for the formation of a quasicrystalline monolayer. It is an interesting result and the authors' analysis is convincing.

Reply: Thank you very much.

#2: Taken on their own, the VASP calculations seem to be a weak point in this study. Some evidence should be provided to support the choice of the calculation cell size and the choice of degrees of freedom to relax. The lateral size of the cell seems small: especially if there is any significant relaxation of surface or subsurface atoms, the slab ought to extend well beyond the fullerene radius if the results are to be used quantitatively.

Reply: We would like to thank the referee for showing us this misleading aspect of our manuscript. Indeed, Figure 3(b) does not depict the unit cell used in our DFT simulations. More precisely, as already mentioned in the Methods section, the VASP calculations for a single C₆₀ molecule on 2Pt-Pt₃Ti(111) were performed using a slab consisting of 5 atomic layers with a (3x3) hexagonal in-plane surface supercell. This implies that the calculated in-plane surface supercell is described by a lattice parameter of 1.675 nm such that the molecule-molecule separation distance is larger than 1 nm.

Changes in the manuscript: To illustrate this point more clearly, we added two images of the supercell used in our simulations with the corresponding figure caption in Supplementary Information (Fig. S4a and Fig. S4b).

#2: *Also, the bridge1 and bridge2 sites are close enough in their calculated adsorption energies that they should probably be regarded as qualitatively isoenergetic (instead of "most highly" and "less" favored) -- even if VASP's accuracy is assumed at this level (dubious), the energy separation is on the order of kT at the surface preparation temperature.*

Reply: Like mentioned in the manuscript (bottom of page 9, top of page 11 and middle of page 12) we fully agree with the referee that bridge 1 and bridge 2 sites are nearly isoenergetic.

Changes in the manuscript: Since the term "most highly favored" (only put there in order to discriminate between the two favored bridge positions) is obviously misleading, we use now only "favored bridge1 position" in the text (on page 12 and 13).

#2: *The authors indicate that a quasicrystalline monolayer also forms when Pt₃Ti has only a single overlayer of platinum. This data should be included, either in the paper or the SI.*

Changes in the manuscript: We have added a corresponding Figure and figure caption to the SI, Fig. S6.

#2: *If at all possible, the authors should repeat their experiment with a Pt(111) surface, as well, to verify that under the specific conditions used, they do not observe quasicrystalline ordering. Adsorption energy calculations for both of these surfaces (Pt₃Ti-1 Pt and Pt(111)) should be done in a similar fashion to the calculations presented. The authors' interesting and surprising conclusion in this study is that third-layer Ti atoms modify the energy landscape enough to drive the formation of a quasicrystalline monolayer. The strongest possible argument to support this conclusion would be to show that the adsorption energies are appropriately modified in all cases where quasicrystals are observed (Pt₃Ti with 2 layers and 1 layer of Pt), and that they are not in adsorption systems (presumably, C₆₀ on Pt(111)) where only hexagonal ordering is seen.*

Reply: C₆₀ has been adsorbed on a pure Pt(111) surface by quite a number of well-known scientists, likewise under comparable deposition conditions, [17, 18] and others. All of them reported that a ($\sqrt{13} \times \sqrt{13}$) R13.9° superstructure of fullerenes forms at room temperature on Pt(111) and is stable up to 420 K. In addition, the most stable adsorption site of a fullerene in this superstructure is supposed to be the bridge position (here exists only one) with an adsorption energy of -4.59 eV [18]. This value corresponds very nicely to our calculation of the adsorption energy in bridge3 position on the 2Pt-Pt₃Ti surface which is -4.543 eV. Note that this is the bridge position between a Pt_{top} and a Pt_{top}Ti. The bridge1 and bridge2 positions are located between two Pt_{top}Ti positions and exhibit higher adsorption energies of -4.796 eV and -4.754 eV, respectively.

On the Pt₃Ti surface covered by only one layer of Pt, again three considerably different bridge positions are available due to geometric reasons. However, these bridge positions should exhibit even higher adsorption energies for fullerenes since they are in closer contact to Ti atoms located already in the next layer. This is in accordance with the smaller size of the quasicrystalline domains resulting on this surface, like can be seen in the Figure added to the SI (Fig. S6). As first approximation, higher adsorption energies cause smaller diffusion length and thus, could lead to smaller quasicrystalline structures, like observed here.

Changes in the manuscript: We added some comments in the manuscript in order to make these aspects more clear to the reader:

- p.13: “However, a ($\sqrt{13} \times \sqrt{13}$) R13.9° superstructure of fullerenes, as is commonly observed on the Pt(111) surface, which exhibits only one kind of bridge positions, can be realized on the 2Pt-Pt₃Ti(111) surface only if alternatively bridge1 and other less favored bridge2 or bridge3 positions are occupied.”
- p.13: “Based on geometrical considerations three different kinds of bridge sites (with respect to subsurface Ti atoms) can be identified on the Pt-Pt₃Ti(111) surface. Here, the subsurface Ti atoms should influence the local adsorption energy of fullerenes even more than in the case of the 2Pt-Pt₃Ti(111) surface since the surface Pt atoms are in direct interaction with the subsurface Ti atoms. This should lead to a considerable energetic discrimination of the three different bridge positions, which in turn results in the formation of a 2D dodecagonal QC, like shown in Fig. S6.”

Reviewers' comments:

Reviewer #1 (Remarks to the Author):

The detailed answer of the authors to the main question if the observed structure is a dodecagonal fullerene quasicrystal does not resolve the issue. The revised manuscript is still not scientifically sound and in several points misleading as I will discuss below. My comments are in black to the answers of the authors (in blue):

The referee argues that the wagon-wheel structure is sufficient to explain the FFT of our experimental STM image and the triangle/square ratio. Indeed, the calculated diffraction pattern of the wagon-wheel structure resembles our FFT. However, the wagon-wheel structure is not the only structure that leads to such a diffraction pattern. There are diverse other structures leading to FFT patterns identical within the experimental resolution, one of which is the phason-strained dodecagonal quasicrystal we claim to have observed. Corroboratively, it has been demonstrated recently that the Fourier spectrum does not suffice to differentiate between a real-space dodecagonal quasicrystalline structure or a related periodic structure [R1].

This answer is not convincing at all. A distinct LEED pattern clearly defines the surface unit cell in the case of a periodic structure or defines the quasicrystalline aperiodic order. There is certainly no ambivalence. The cited reference by Tito et al. does NOT show that “that the Fourier spectrum does not suffice to differentiate between a real-space dodecagonal quasicrystalline structure or a related periodic structure”. It deals with nonperiodic structures.

In Fig. 2 of his report, the referee compares the calculated pattern of a hexagonal wagon-wheel arrangement with the FFT of our experimental STM image. The referee does so by using a grid superposed to the experimental pattern. The peaks in the FFT indeed correspond to crossing points of that grid. Upon careful inspection, however, it occurs that the grid used by the referee does not consist of equidistant lines as it should for a periodic pattern. It is clearly visible that the distances between the grid lines have been adjusted such that the crossing points match the peaks of the FFT. This procedure is highly suggestive and appears questionable to us. Using, on the other hand, a pattern of equidistant grid lines, a perfect match of all peaks of the FFT with crossing points is not possible (Fig. R1a).

Any STM image is subject to distortions and a time-dependent creep due to the piezo scanners. Therefore, one has to be careful to undistort the image, which is similarly effecting the FFT of the STM image. Therefore, the small deviations of the peak positions from a perfect grid of equidistant lines is a difficult argument. The authors should have inspected the original STM data. By doing so, they would have found several patches of the periodic grid also in real space, despite the disorder in the overall structure. On the other hand, a careful inspection of the real space data does not show any dodecagonal structure element on the $2+\sqrt{3}$ larger length scale, again ruling out aperiodic order.

Furthermore, an (equidistant!) hexagonal grid can be used to describe the peak maxima of Fig. 14a in the paper of Socolar (Ref [25]), with a similar matching agreement as to our FFT pattern (Fig. R2). Notably, the figure of Socolar is the diffraction pattern of a dodecagonal quasicrystal including phason strain, i.e. exactly the type of structure we claim to have observed in our experimental images.

No! The diffraction pattern of Socolar can easily be generated by linear combination of four (30° rotated) unit vectors. The linear combination of only two (60° rotated with respect to each other, no phason strain) unit vectors produces a hexagonal grid. However taking the other two vectors also into account yields a dense nonperiodic array of diffraction maxima. This point holds also in the presence of linear phason strain.

We thus conclude that our experimental FFT pattern does neither allow to conclude on the presence of periodic hexagonal structure, nor rule out the presence of a phason-strained dodecagonal quasicrystal.

NO, see argumentation above. All presented data are well described by a disordered square-triangle, but periodic structure.

The referee states that the triangle/square ratio of our structure matches that of a hexagonal wagon-wheel arrangement. However, it has been demonstrated that the presence of a given triangle/square ratio does not allow to conclude on the real-space structure (which simply means that different structures can be constructed using a given ratio of triangle and square tiles). It has, for example, been demonstrated that random-Stampfli approximants at zero phason strain possess exactly the same triangle/square ratio of 2.309 [R1] as deterministic (i.e. ideal) and random dodecagonal tilings [9].

I perfectly agree that a deterministic dodecagonal tiling and a random or disordered dodecagonal tiling share the same triangle/square ratio. However, this ratio is different (and especially rational) for a periodic structure. Therefore, the experimentally observed ratio fits nicely to the proposed periodic structure, but disagrees with the value of a quasicrystal.

Finally, the referee's statement, saying that the periodic hexagonal wagon-wheel structure "can explain straightforwardly all experimental observations" is not true. Factually our real-space STM image (Fig. 2a) and its tiling (Fig. 2b) are in direct contradiction with the mere presence of a periodic hexagonal wagon-wheel structure. The structure in the experimental STM image is clearly not identical to the referee's Fig. 1b representing the wagon-wheel structure. The surface section in our experimental image is composed of small local areas, which individually can be identified as approximants of the ideal dodecagonal quasicrystal. It contains 8/3 areas, the approximant corresponding to the hexagonal periodic wagon-wheel structure, but it also contains areas of 32.4.3.4 approximant, hexagonal approximant, consecutive squares and diverse transition regions.

The authors should analyze their data in more depth. It is not the duty of the referee to work out of complete new analysis. However, in the attached pdf file I highlighted some patches where the periodic structure prevails (in the attached graph, the centers of 3⁶ structures that belong to a single, strictly periodic structure are highlighted). Clearly, this periodic phase does contribute to a substantial fraction (if not all) of coherent scattering. The FFT shows the diffraction pattern of exactly this periodic structure. Note further, the fixed 60° alignment of all these patches.

Generally, a quasicrystalline structure, even an ideal one, can always locally be decomposed into small areas which individually can be identified as characteristic tiling patches corresponding to approximants of different order [2]. Nevertheless, the total structure still remains a quasicrystal, unless the local approximant areas are periodically repeated on a nonlocal scale.

The statement "Nevertheless, the total structure still remains a quasicrystal, unless the local approximant areas are periodically repeated on a nonlocal scale" shows a misconception: A missing periodic order is not what defines aperiodic order; instead, there needs to be a long-range order (e.g. on the basis of self-similarity or due to projection from higher-dimension space) to qualify for a quasicrystal. More specifically for a disordered system like here, it is the coherence between patches which decides if the structure is long-range periodic, aperiodic or disordered.

We also do not agree with the referee's statement that our data does not provide evidence for the presence of phason strain. Following the procedure described by Leung et al. [26] for the identification of phason strain in a dodecagonal quasicrystal, we have obtained the nonzero phason matrix $1.3 \begin{pmatrix} 1 & 0.1 \\ -0.1 & -1 \end{pmatrix}$. This phason matrix corresponds to the presence of Γ^{1b} phason strain in a dodecagonal quasicrystal [25]. Note furthermore that our matrix is significantly different from the phason matrix corresponding to an 8/3 approximant. The latter reads $0.26 \begin{pmatrix} 1 & -0.3 \\ 0.3 & 1 \end{pmatrix}$ [11], which further supports our statements above.

These points demonstrate that the structure observed experimentally can be described as a dodecagonal quasicrystal containing phason strain. The argumentation line of the referee, who states that the structure observed corresponds to a wagon-wheel structure or 8/3 approximant, is not conclusive and partly in conflict with the experimental facts.

It is known that a (artificially) large linear "phason strain" can transform a quasicrystalline structure into a periodic one, the so-called approximant structures. Therefore, the

assignment of a nonzero phason matrix does not allow concluding on aperiodic order. Instead, a large value for the linear phason strain might even question an aperiodic structure in general. Note that the reported phason strain is by far larger than for known quasicrystals.

The quasicrystalline domains are dominant in the fullerene monolayer, but the hexagonal phase still constitutes 30% to 40% of the surface area. Accordingly, peaks of hexagonal domains are present in the LEED pattern. To make this issue more clear we included two images in the SI (Fig. S1a and Fig. S1b).

In the revised manuscript is the statement “quasicrystalline domains cover 60 % to 70 % of the surface area.” is not supported by the data as discussed above. It remains fully unclear on which basis the author judge the quasicrystalline order.

I do not recommend publication.

Reviewer #2 (Remarks to the Author):

As stated in the first round of review, I believe that this paper describes an interesting result, and that the authors' analysis is convincing. To elaborate upon the latter point, I believe the authors are correct in describing their observed structure as quasicrystalline. In particular, the real-space images in figures 1 and 2 show structures that are clearly not periodic, and the FFTs are sharp and show angular symmetry disallowed in periodic lattices -- these are the necessary and sufficient conditions set out by the IUC (International Union of Crystallographers) for quasicrystallinity.

I support publication of this manuscript in its current form. The authors could potentially make the study stronger by addressing the following:

- 1) I continue to be concerned about the lateral size of the unit cell used for the VASP calculations. The authors' response is that the C60-C60 separation is larger than 1 nm. This is sufficient to rule out the relatively short-range forces caused by direct intermolecular interaction; however, indirect interactions, where the adsorbate modifies the surface structure, are not necessarily nearly so local. I suspect that the arrangement of surface and sub-surface atoms will not be independent of cell size until the box is substantially larger, and so the fairly subtle differences in adsorption site energies are potentially not accurate. The authors' argument that the box is large enough may or may not be correct; however, repeating a calculation with a larger box size and showing essentially unchanged numbers would make a very strong argument.
- 2) The authors argue that C60 on Pt(111) is well covered in the literature, both through STM and electronic-structure calculations. They are correct that for this reason, a new measurement is not required. However, there is nothing quite as convincing as a direct comparison made side-by-side on the same instrument and under the same conditions, and I encourage the authors to repeat the C60/Pt(111) experiment for this reason.

Response to the reviewers comments from Nov. 17th 2016 on

Manuscript: NCOMMS-16-16737B-Z

“Interface-driven formation of a two-dimensional dodecagonal fullerene quasicrystal”

We thank the reviewers again for their comments. In the following, we reply to all questions raised, and address all subjects of debate. In his current set of comments, reviewer #1 has quoted our initial reply. For the sake of clarity, we here represent the reviewer’s comments in boxes, which are immediately followed by our current reply (in green).

Citations quoted as numbers in square brackets in this reply refer to the references in the manuscript. Additional citations used only here are quoted as [Rx]. Analogously we refer to figures in the manuscript and in the reply.

Reply to the comments of reviewer #1:

Reviewer #1: The detailed answer of the authors to the main question if the observed structure is a dodecagonal fullerene quasicrystal does not resolve the issue. The revised manuscript is still not scientifically sound and in several points misleading as I will discuss below. My comments are in black to the answers of the authors (in blue):

Authors 1. reply (in blue): The referee argues that the wagon-wheel structure is sufficient to explain the FFT of our experimental STM image and the triangle/square ratio. Indeed, the calculated diffraction pattern of the wagon-wheel structure resembles our FFT. However, the wagon-wheel structure is not the only structure that leads to such a diffraction pattern. There are diverse other structures leading to FFT patterns identical within the experimental resolution, one of which is the phason-strained dodecagonal quasicrystal we claim to have observed. Corroboratively, it has been demonstrated recently that the Fourier spectrum does not suffice to differentiate between a real-space dodecagonal quasicrystalline structure or a related periodic structure [R1].

Reviewer #1 (in black): This answer is not convincing at all. A distinct LEED pattern clearly defines the surface unit cell in the case of a periodic structure or defines the quasicrystalline aperiodic order. There is certainly no ambivalence. The cited reference by Tito et al. does NOT show that “that the Fourier spectrum does not suffice to differentiate between a real-space dodecagonal quasicrystalline structure or a related periodic structure”. It deals with nonperiodic structures.

Authors 2. reply: In the quoted reference [R1] Zito et al. consider a quasiperiodic triangle-square tiling generated with Stampfli’s inflation procedure. The authors show that this nonperiodic structure can be reconstructed by means of a suitably chosen supercell repeated in a triangular periodic lattice, and the Fourier spectrum of the latter exhibits 12-fold rotational symmetry. Consequently, in the case of dodecagonal quasicrystals the diffraction pattern does not suffice to differentiate between a real space dodecagonal quasicrystalline structure and a related periodic structure, as we wrote in our initial reply.

This however, is not the critical point. In our reply we have demonstrated that the FFT of our experimental real space structure equally well fits that of a phason strained dodecagonal quasicrystal (constructed by Socolar et al., Ref. [25], Fig. R1c)) and that of a wagon-wheel structure (Fig. R1b). This means that the FFT is consistent with the presence of a phason-strained dodecagonal quasicrystal in our experiments, but it is not unambiguous evidence. On the other hand, however, it also shows that our FFT does not allow for the conclusion that our structure is a periodic wagon-wheel structure, as claimed by the reviewer.

Fig. R1: a) FFT of our STM image (manuscript Fig. 2f) with overlaid reviewer grid (green) and equidistant hexagonal grid (blue). The equidistant grid obviously deviates from the non-equidistant grid (green) used by the referee. b) Ideal diffraction pattern of the wagon-wheel structure. c) Equidistant hexagonal grid (blue) superposed to the diffraction pattern of a dodecagonal quasicrystal including phason strain (Fig. 14 of Socolar [25]).

Authors 1. reply: In Fig. 2 of his report, the referee compares the calculated pattern of a hexagonal wagon-wheel arrangement with the FFT of our experimental STM image. The referee does so by using a grid superposed to the experimental pattern. The peaks in the FFT indeed correspond to crossing points of that grid. Upon careful inspection, however, it occurs that the grid used by the referee does not consist of equidistant lines as it should for a periodic pattern. It is clearly visible that the distances between the grid lines have been adjusted such that the crossing points match the peaks of the FFT. This procedure is highly suggestive and appears questionable to us. Using, on the other hand, a pattern of equidistant grid lines, a perfect match of all peaks of the FFT with crossing points is not possible (Fig. R1a).

Reviewer #1: Any STM image is subject to distortions and a time-dependent creep due to the piezo scanners. Therefore, one has to be careful to undistort the image, which is similarly affecting the FFT of the STM image. Therefore, the small deviations of the peak positions from a perfect grid of equidistant lines is a difficult argument. The authors should have inspected the original STM data. By doing so, they would have found several patches of the periodic grid also in real space, despite the disorder in the overall structure. On the other hand, a careful inspection of the real space data does not show any dodecagonal structure element on the $2+\sqrt{3}$ larger length scale, again ruling out aperiodic order.

Authors 2. reply:

Indeed, it has to be taken into account that STM images may be distorted, and in that case the FFT has to be equalized before interpretation. However, such rectification has to be done by careful comparison of experimental and ideal spot positions for a well-known reference sample

and can certainly not be done by arbitrary displacement of grid lines. Fig. R1a, comparing reviewer's grid with a true equidistant grid (blue), demonstrates that the reviewer's grid displays irregular deviations, clearly not corresponding to typical distortions caused e.g. by creep of the piezo scanners. These deviations obviously have been introduced to achieve a better fit to support the reviewer's statements.

We have of course checked whether or not rectification of our experimental images is required and came to the conclusion that this is not the case. Our LT-UHV-STM is extremely stable (the drift at 77 K is smaller than 3 pm/min, while the scan time for one image is 5 min and the fullerene-fullerene distance is about 1 nm).

In the revised manuscript, we already included a real-space analysis of our original STM data (page 6, Fig. 2b). The tiling representation indeed reveals the presence of 8/3 approximant, but also areas of 3.3.4.3.4 approximant, hexagonal approximant, consecutive squares and diverse transition regions. This finding is in full agreement with the presence of a dodecagonal quasicrystal. As we wrote in our first rebuttal: "Generally, a quasicrystalline structure, even an ideal one, can always **LOCALLY** be decomposed into small areas which individually can be identified as characteristic tiling patches corresponding to approximants of different order [2]. Nevertheless, the total structure still remains a quasicrystal, unless the local approximant areas are periodically repeated on a nonlocal scale."

Finally, the lack of higher-order dodecagonal structure on a larger length scale is **NOT** a condition for the presence of a dodecagonal quasicrystal. In the case of a random-tiling quasicrystal, a higher-order dodecagonal structure is not expected [1], [R2]. Accordingly, we have clearly stated in our manuscript (page 7, line 5):

"We do not find a higher-order tiling, i.e. a larger tiling that is inflated by a factor of $(2 + \sqrt{3})$, and thus conclude that the present dodecagonal QC corresponds to a random tiling."

Authors 1. reply: Furthermore, an (equidistant!) hexagonal grid can be used to describe the peak maxima of Fig. 14a in the paper of Socolar (Ref [25]), with a similar matching agreement as to our FFT pattern (Fig. R2). Notably, the figure of Socolar is the diffraction pattern of a dodecagonal quasicrystal including phason strain, i.e. exactly the type of structure we claim to have observed in our experimental images.

Reviewer #1: No! The diffraction pattern of Socolar can easily be generated by linear combination of four (30° rotated) unit vectors. The linear combination of only two (60° rotated with respect to each other, no phason strain) unit vectors produces a hexagonal grid. However taking the other two vectors also into account yields a dense nonperiodic array of diffraction maxima. This point holds also in the presence of linear phason strain.

Authors 2. reply: For our argumentation, it does not matter how the diffraction pattern of Socolar **CAN** be created. It is a fact, clearly stated in the reference [25], that the diffraction pattern under consideration **IS** that of a dodecagonal quasicrystal including phason strain. We claim that our experimentally observed structure is a phason-strained quasicrystal as well, and

therefore it seems adequate to us that we compare the FFT of our experimental STM image with this diffraction pattern.

Authors 1. reply: We thus conclude that our experimental FFT pattern does neither allow to conclude on the presence of periodic hexagonal structure, nor rule out the presence of a phason-strained dodecagonal quasicrystal.

Reviewer #1: NO, see argumentation above. All presented data are well described by a disordered square- triangle, but periodic structure.

Authors 2. reply: In addition to our reply above we would like to repeat our reply provided in the first rebuttal:

“Finally, the reviewers’ statement, saying that the periodic hexagonal wagon-wheel structure “can explain straightforwardly all experimental observations” is not true. Factually our real-space STM image (Fig. 2a) and its tiling (Fig. 2b) are in direct contradiction with the mere presence of a periodic hexagonal wagon-wheel structure. The structure in the experimental STM image is clearly NOT identical to the reviewers Fig. 1b representing the wagon-wheel structure.”

Authors 1. reply: The referee states that the triangle/square ratio of our structure matches that of a hexagonal wagon- wheel arrangement. However, it has been demonstrated that the presence of a given triangle/square ratio does not allow to conclude on the real-space structure (which simply means that different structures can be constructed using a given ratio of triangle and square tiles). It has, for example, been demonstrated that random-Stampfli approximants at zero phason strain possess exactly the same triangle/square ratio of 2.309 [R1] as deterministic (i.e. ideal) and random dodecagonal tilings [9].

Reviewer #1: I perfectly agree that a deterministic dodecagonal tiling and a random or disordered dodecagonal tiling share the same triangle/square ratio. However, this ratio is different (and especially rational) for a periodic structure. Therefore, the experimentally observed ratio fits nicely to the proposed periodic structure, but disagrees with the value of a quasicrystal.

Authors 2. reply: As we have demonstrated in our first rebuttal, there exists no one-to-one relation between the triangle/square ratio and the underlying structure. Furthermore it is questionable if it makes sense to discuss whether the experimentally determined ratio of 2.67 corresponds to a rational number or not.

Authors 2. reply: First of all, we have included a real-space analysis of our STM images in the revised version (p.6, Fig. 2b and p.7), which demonstrates the presence of diverse types of approximant. We also included a statement saying that we indeed observe a slight prevalence of the 8/3 approximant (p.9).

Fig. R2 is the figure referred to by the reviewer. Indeed it contains parts covered by 3^6 structures (highlighted by the reviewer in red – however, a considerable amount of the highlighted

structures had to be cancelled by us with blue bars, since they are no 3^6 structures), but the corresponding surface area obviously is a minor fraction of the total area, which roughly corresponds to that in our analysis in Fig. 2b. Therefore, it is indefensible to claim that this phase dominates the formation of the diffraction pattern.

Fig. R2: 3^6 centers are highlighted in red by reviewer #1. Blue bars cancel wrongly assigned 3^6 centers.

Authors 2. reply: Indeed from a lack of periodicity it cannot be concluded that aperiodic order exists. However, we do not argue that naïve. The original statement of Steurer and Deloudi ([2] p. 324) reads:

“Everywhere in an ideal QC, portions of approximants of any order exist locally. The strict order of the QC is related to the stability of all this local realizations of approximants. The structure can be phason-strained if one of the approximants wins the stability race.”

This passage shows that the local occurrence of approximants of different type is characteristic and a generic property of quasicrystals. Our experimental observations are in perfect accordance with this statement, and hence the presence of whatever type of approximant in our experimental structures cannot be interpreted as an indication that our structure is a periodic approximant.

In further agreement with the statement of Steurer and Deloudi, we find a slight prevalence of the $8/3$ approximant (as stated on p. 9), which is consistent with our observation of the presence of phason strain.

Authors 2. reply: Phason strain mediates a continuous transition from ideal quasicrystals to low-order approximants with small lattice constants. This is a well-established understanding in the quasicrystal community, see e.g. Ishii [R3]. Accordingly, there exists no concept of a limiting phason strain above which a structure necessarily is considered a periodic approximant, as the reviewer seems to suggest.

The reported phason strain is larger by less than a factor of two, than that discussed by Socolar [25] in his analysis of phason strained dodecagonal quasicrystals (a prefactor of 1.3 here vs. 0,71 in Socolar’s analysis), and thus, can by no means be considered prohibitively large.

Authors 2. reply: We find two different surface structures formed by the fullerenes on the $2\text{Pt-Pt}_3\text{Ti}(111)$. Besides the dodecagonal quasicrystal, identified according to the analysis described in our manuscript, we find a hexagonal structure. The latter is denser, periodic and can easily be distinguished from the quasicrystalline areas (see Fig. 1 and the lower right corner in Fig. R2). In total, we have analyzed 4 STM images and measured the respective surface area of the two phases. As a result, we find that on average about 60 to 70 % of the surface is covered by the

quasicrystalline structure. Fig. S1 in the supplementary information file represents a typical surface section in this respect.

We have added a corresponding statement on p. 3:

“In the resulting fullerene monolayer, we find quasicrystalline and hexagonal domains. The analysis of four STM images revealed that on average the quasicrystalline domains cover 60 % to 70 % of the surface area.”

Reply to the comments of reviewer #2 (in green):

Reviewer #2 (Remarks to the Author):

As stated in the first round of review, I believe that this paper describes an interesting result, and that the authors' analysis is convincing. To elaborate upon the latter point, I believe the authors are correct in describing their observed structure as quasicrystalline. In particular, the real-space images in figures 1 and 2 show structures that are clearly not periodic, and the FFTs are sharp and show angular symmetry disallowed in periodic lattices -- these are the necessary and sufficient conditions set out by the IUC (International Union of Crystallographers) for quasicrystallinity.

I support publication of this manuscript in its current form.

Authors 2. reply: Thank you very much!

The authors could potentially make the study stronger by addressing the following:

- 1) I continue to be concerned about the lateral size of the unit cell used for the VASP calculations. The authors' response is that the C60-C60 separation is larger than 1 nm. This is sufficient to rule out the relatively short-range forces caused by direct intermolecular interaction; however, indirect interactions, where the adsorbate modifies the surface structure, are not necessarily nearly so local. I suspect that the arrangement of surface and sub-surface atoms will not be independent of cell size until the box is substantially larger, and so the fairly subtle differences in adsorption site energies are potentially not accurate. The authors' argument that the box is large enough may or may not be correct; however, repeating a calculation with a larger box size and showing essentially unchanged numbers would make a very strong argument.
- 2) The authors argue that C60 on Pt(111) is well covered in the literature, both through STM and electronic-structure calculations. They are correct that for this reason, a new measurement is not required. However, there is nothing quite as convincing as a direct comparison made side-by-side on the same instrument and under the same conditions, and I encourage the authors to repeat the C60/Pt(111) experiment for this reason.

Authors 2. reply: We thank the reviewer for these valuable comments.

[R1] Zito, G., Pepe, G. P. & De Nicola, S. Hidden translational symmetry in square-triangle-tiled dodecagonal quasicrystal. *J. Opt.* **17**, 055103 (2015).

[R2] DiVincenzo, D. P., Steinhardt, P. J., Series on Directions in Condensed Matter Physics – Vol.16, Quasicrystals, The state of the art, World Scientific, Singapore 1999, page 475ff.

[R3] Ishii, Y. Mode locking in quasicrystals. *Phys. Rev. B* **39**, 11862 (1989).

Reviewers' comments:

Reviewer #4 (Remarks to the Author):

I have read the manuscript and supplemental material as well as the discourse between the authors and Reviewer 1. The crux of the matter seems to be a disagreement regarding whether the self-assembled monolayer of fullerenes can be characterized as quasicrystalline or, rather, a collection of periodic crystals with motifs that exhibit local 12-fold symmetry.

The author's claims are based upon STM and Fourier transforms of the STM data. In Figure 1, they show that the purported QC coexists with a hexagonal phase and state that the QC phase occupies approximately 60-70% of the surface. There are clearly multiple defect sites in the QC, whereas the hexagonal phase is rather "well-behaved". In Figure 2, the authors focus on the QC region and show that the QC can, at least locally, be subdivided into smaller regions corresponding to approximants of the dodecagonal quasicrystal along with some defects. Indeed, on page 7 they state that "The dodecagonal QC can be decomposed into different local areas corresponding to different types of approximants.

The question is whether this collection of approximants, in total, can be considered to be a true quasicrystal, albeit heavily defected and containing significant phason strain. In other words, when does one cross the boundary from a QC to a collection of periodic approximants. As far as I know, this question has not been answered quantitatively in the literature and is quite interesting in itself.

Certainly, on small enough length scales, the difference between a QC and a large unit cell approximant loses meaning. In my opinion the real question is what the size of each approximant patch (in Figure 2b for example) relative to, say, a 2D unit cell of the approximant. It is very difficult to discern this from the data that has been presented. Some information regarding these length scales would be very helpful.

As it stands right now, I am hard pressed to say that the authors have provided convincing evidence that this can be considered a true QC, random tiling or not, although the paper itself is quite interesting.

Response to the Reviewer #4

Manuscript: NCOMMS-16-16737C

“Interface-driven formation of a two-dimensional dodecagonal fullerene quasicrystal”

We thank the reviewer for his comments. In the following, we reply to all questions raised, and address all subjects of debate.

Reviewer #4 (Remarks to the Author):

I have read the manuscript and supplemental material as well as the discourse between the authors and Reviewer 1. The crux of the matter seems to be a disagreement regarding whether the self-assembled monolayer of fullerenes can be characterized as quasicrystalline or, rather, a collection of periodic crystals with motifs that exhibit local 12-fold symmetry.

Authors reply: This is the correct description of the discussion between Reviewer #1 and us. However, we would like to add that Reviewer #2 proved true that our observed structure is quasicrystalline and clearly not periodic, and even stated that the necessary conditions for quasicrystallinity set out by the IUC (International Union of Crystallographers) are met by our structures.

Furthermore, we would like to point out that the main message of our manuscript is not about quasicrystallinity but about the mechanisms that lead to specific triangle-square tilings. This new message, however, might bring the scientific community a step closer to the answer of the question ‘Why do quasicrystals form?’

The author's claims are based upon STM and Fourier transforms of the STM data. In Figure 1, they show that the purported QC coexists with a hexagonal phase and state that the QC phase occupies approximately 60-70% of the surface. There are clearly multiple defect sites in the QC, whereas the hexagonal phase is rather "well-behaved". In Figure 2, the authors focus on the QC region and show that the QC can, at least locally, be subdivided into smaller regions corresponding to approximants of the dodecagonal quasicrystal along with some defects. Indeed, on page 7 they state that "The dodecagonal QC can be decomposed into different local areas corresponding to different types of approximants.

The question is whether this collection of approximants, in total, can be considered to be a true quasicrystal, albeit heavily defected and containing significant phason strain. In other words, when does one cross the boundary from a QC to a collection of periodic approximants. As far as I know, this question has not been answered quantitatively in the literature and is quite interesting in itself.

Authors reply:

The necessary and sufficient conditions, quoted by reviewer #2 as a definition of quasicrystallinity, are i) the absence of real-space periodicity, ii) the presence of an essentially discrete diffraction pattern, and iii) the presence of a symmetry axis that is forbidden in periodic crystals. It is still under debate (see R. Lifshitz, arXiv:cond-mat/0008152v1) whether or not the third criterion, should be included in the definition, since i) and ii) are sufficient. However, the structure described and discussed in our manuscript clearly fulfills all three criteria: The real-space images of the structure labeled “QC” in Figs. 1 and Fig. 2 are not periodic. Their FFT displays discrete spots and has a symmetry that is forbidden in periodic crystals. We hence feel safe in our conclusion that our fullerene-based structure is a two-dimensional quasicrystal.

We agree with the assumption of reviewer #4 that there exists no limiting degree of phason distortion, above which a QC necessarily has to be considered a periodic crystal. Rather, it has been demonstrated (Ishii, Y, Phys. Rev. B 39, 11862 (1989)) that phason strain mediates a continuous transition from ideal quasicrystals to low-order approximants, down to very small lattice constants. Our quantitative analysis demonstrates that the phason strain is larger than that discussed by Socolar [25] in his treatment of phason strained dodecagonal quasicrystals by less than a factor of two. We find a slight prevalence of the 8/3 approximant (as stated on p. 9), which is consistent with the presence of phason strain. This demonstrates that there is indeed a significant amount of phason strain in our structures, but it is not overly large and still allows the interpretation in terms of a distorted quasicrystal.

In the QC phase a higher density of defects is present in comparison to the hexagonal phase. However, this is commonly observed and characteristic for QC phases. For example, the defect density in Fig. R1b (Fig. 3 in Hayashida et. al. [10]) is comparable to ours, Fig. R1a (Fig. 2b of our manuscript). The defects in Fig. R1a are polygons with interior angles of 60° , 90° , 120° and 150° consistent with the square-triangle tiling and point to the fact that the QC is not in its ground state [Dotera, Nature, **2014**, 506, 208].

Fig. R1a: Triangle-square tiling with color coded decomposition into approximants (= Fig. 2b of our manuscript)

8/3 Approx.:	37%
$3^2.4.3.4$ Approx.:	8%
3^6 Approx.:	10%
4^2 :	11%
Transition region:	30%
Defects:	4%

Fig. R1b: Triangle-square tiling with color coded decomposition into approximants (= Fig. 3 in Hayashida et. al. [10])

8/3 Approx.:	27% (yellow and green)
$3^2.4.3.4$ Approx.:	17%
3^6 Approx.:	0%
4^2 :	3%
Transition region:	48%
Defects:	5%

The comparison of the tiling describing our structure (Fig. R1a) with that of Hayashida et. al [10] (Fig. R1b) displays mainly quantitative differences. The same types of approximant (except for 3^6) are present at comparable number distributions.

Action (page 7): We added the relative amount of area covered by the different approximants and included the following sentence:

“The local occurrence of approximants of different types is a characteristic and generic property of quasicrystals and our experimental observations are in perfect accordance with this feature.”

Certainly, on small enough length scales, the difference between a QC and a large unit cell approximant loses meaning. In my opinion the real question is what the size of each

approximant patch (in Figure 2b for example) relative to, say, a 2D unit cell of the approximant. It is very difficult to discern this from the data that has been presented. Some information regarding these length scales would be very helpful.

Authors reply:

On a small length scale, necessarily the differentiation between QCs and approximants breaks down, and hence local portions of structure can always be classified as approximants. Still, the ensemble of different approximants together makes up a quasicrystal. This is clearly stated in the monograph “The Crystallography of Quasicrystals” by Steurer and Deloudi ([2], p. 324): “Everywhere in an ideal QC, portions of approximants of any order exist locally. The strict order of the QC is related to the stability of all this local realizations of approximants. The structure can be phason-strained if one of the approximants wins the stability race.” This passage reflects the understanding, well established in the community, that the local occurrence of approximants of different type is characteristic and a generic property of quasicrystals.

The relative areas covered by different types of approximant are given in the figure caption of Fig. R1 for our triangle-square tiling in direct comparison with an approved example from the literature [10]. The 8/3 approximant covers the largest area in both cases. The remaining area (that is in our case about 60% of the area) consists of different approximants and transition regions (which are not defects). Indeed, experimentally the question how many unit cells of an approximant are comprised in a given patch is difficult to answer. The result depends on the way of counting (only complete unit cells and/or overlapping unit cells?). For example, we find in our largest 8/3 approximant patch 3 to 5 unit cells (Fig. R1a), while we find in Fig. R1b 3 to 8 unit cells in the largest 8/3 approximant patch, due to more overlapping unit cells.

Nevertheless, the entirety of all areas together, in consistence with the statement from Steurer and Deloudis book, is a QC fulfilling all criteria i) to iii) quoted above of the definition.

As it stands right now, I am hard pressed to say that the authors have provided convincing evidence that this can be considered a true QC, random tiling or not, although the paper itself is quite interesting.

Author reply: We hope, that we could convince the reviewer, that our two-dimensional dodecagonal fullerene QC meets all necessary requirements of a QC and exhibits features comparable to other quasicrystalline tilings given in literature.

Reviewers' comments:

Reviewer #4 (Remarks to the Author):

The authors have provided a reasonable response to my concerns about the manuscript. Their argument, however, seems to rest on the monograph by Steurer and Deloudi and, in particular, on the meaning of the statement: "Everywhere in an ideal QC, portions of approximants of any order exist locally. The strict order of the QC is related to the stability of all this local realizations of approximants. The structure can be phason-strained if one of the approximants wins the stability race."

What they show in the figure included in their response (Fig. R1) is quite clearly a collection of approximant phases separated by transition regions and containing defects. It is not clear to me that this picture is consistent with the description above. The previous referee does not believe this structure to be a true quasicrystal, and I have my doubts. Nevertheless, this is the type of general discussion which the field needs to confront at some point. Therefore, I would not stand in the way of publication of the manuscript.